# Sparse Persistent RNNs: Squeezing Large Recurrent Networks On-Chip

**Feiwen Zhu**[*], **Jeff Pool**[*], **Michael Andersch, Jeremy Appleyard & Fung Xie**
NVIDIA
{mzhu,jpool,mandersch,jappleyard,ftse}@nvidia.com

## Abstract

Recurrent Neural Networks (RNNs) are powerful tools for solving sequence-based problems, but their efficacy and execution time are dependent on the size of the network. Following recent work in simplifying these networks with model pruning and a novel mapping of work onto GPUs, we design an efficient implementation for sparse RNNs. We investigate several optimizations and tradeoffs: Lamport timestamps, wide memory loads, and a bank-aware weight layout. With these optimizations, we achieve speedups of over $6\times$ over the next best algorithm for a hidden layer of size 2304, batch size of 4, and a density of 30%. Further, our technique allows for models of over $5\times$ the size to fit on a GPU for a speedup of $2\times$, enabling larger networks to help advance the state-of-the-art. We perform case studies on NMT and speech recognition tasks in the appendix, accelerating their recurrent layers by up to $3\times$.

## 1 Introduction

Many sequence-based problems, including Speech Recognition  (Amodei et al., 2015) and Neural Machine Translation (NMT)  (Bahdanau et al., 2014), can be solved effectively with Recurrent Neural Networks (RNNs).  Appleyard et al. (2016) showed that these networks can run efficiently on massively parallel processors such as GPUs, and  Diamos et al. (2016) found that if the network is small enough to fit in the register file of a GPU, a persistent approach can be used to increase performance.

In parallel, many network compression methods  (Han et al., 2016c; 2015; Guo et al., 2016) have been shown to reduce the model size of both Convolutional Neural Networks (CNNs) and RNNs. Recent work in this area has found that model pruning, in particular, can lead to significant reductions in the number of important network parameters for RNNs  (Narang et al., 2017; See et al., 2016)

We present an approach that combines both of these techniques into an efficient and expandable approach. In particular, our work makes the following contributions:

- Larger sparse RNNs can be run more efficiently on GPUs.

- Sparse RNNs can be run with smaller batch sizes more efficiently on GPUs.

- Various optimizations on top of the naïve implementation, necessary to achieve high performance.

- Case studies using our technique showing 1) generalization to LSTMs, 2) practical network design considerations, and 3) speedups of up to $3\times$ on two non-synthetic workloads.

A naïve implementation of the idea, presented in Section 3, leads to limited benefit; we present a series of optimizations in Section 4 that help to achieve a high level of performance. Section 5 describes the experimental setup and results, and we discuss future work and our conclusions in Sections 6 and 7. The appendix presents a case study on a machine translation task.

---

[*]Indicates equal contribution

## 2 RELATED WORK

### 2.1 RNNS

Recurrent Neural Networks (RNNs) are powerful tools for solving series-based problems, such as NMT (Bahdanau et al., 2014; Hannun et al., 2014; Amodei et al., 2015; Luong et al., 2015), language modeling (Mikolov et al., 2010), and various NLP tasks (Collobert et al., 2011). More complex recurrent networks have been devised to build on the basic RNN structure, such as Long/Short Term Memory networks (LSTMs) (Hochreiter & Schmidhuber, 1997) and Gated Recurrent Units (GRUs) (Chung et al., 2014). Though we focus on RNNs in this work for simplicity, our approach can extend to other recurrent network types, such as the LSTMs used in our case study. In particular, we build on a recent GPU-based method storing recurrent weights on-chip (Diamos et al., 2016).

### 2.2 MODEL SIMPLIFICATION

Since neural networks are tasked with solving incredibly complex problems, they can become incredibly complex models. So, considerable effort is spent simplifying network models so they run more efficiently and can be deployed on smaller hardware. In this work, we focus on pruning, though other orthogonal simplification techniques (such as quantization (Gong et al., 2014), weight sharing (Chen et al., 2015), and tensor approximations (Cai et al., 2014)) are also possible.

Network pruning is the process of inducing sparsity in the weights of the network model (LeCun et al., 1990; Hassibi et al., 1993) and can be used as a regularizer (Thodberg, 1991; Giles & Omlin, 1994; Han et al., 2017), a method of compression (Han et al., 2015; 2016c), or a way to reduce the computational workload (Han et al., 2016b). Recent results show that this technique is applicable to recurrent networks used in a variety of tasks, from speech recognition (Han et al., 2016a; Narang et al., 2017) to machine translation (See et al., 2016) and image captioning (Han et al., 2016b).

Different pruning techniques include fine-grained, unstructured pruning (Han et al., 2015); regular, structured pruning at a small scale (Anwar et al., 2015; Li et al., 2016; Wen et al., 2016); and pruning entire filters that results in a dense workload with smaller dimensions (Molchanov et al., 2017). In this work, we assume the sparsity is unstructured so as to remain useful in the general case; there are some simplifications that could be made if structure is guaranteed.

## 3 IMPLEMENTATION DETAILS

We draw heavily from Diamos et al. (2016), in which the authors use the large on-chip storage of GPUs to hold the recurrent weights. This approach is briefly discussed here, followed by details of how our sparse technique differs.

### 3.1 RNN OPERATION

A recurrent network's operation is conceptually simple, and each time step can be expressed by Equation 1:

$$h_t = g\left(U_r h_{t-1} + W x_t + b\right) \tag{1}$$

where $U_r$ is the recurrent weight matrix, $W$ is the input-to-hidden weight matrix, $b$ is a bias term, and $g$ is an elementwise activation function. The input-to-hidden weight matrix ($W x_t$) calculation has no dependency, so it can be processed in parallel and added to $b$, becoming $b'$. The formula simplifies to Equation 2:

$$h_t = g\left(U_r h_{t-1} + b'\right) \tag{2}$$

### 3.2 PERSISTENT RNNS

In a persistent RNN implementation, weights $U_r$ are stored in on-chip register files so each thread keeps $x \times y$ ($rows \times columns$) weights, while the activations ($h_{t-1}$) are stored in shared memory. Each row is processed by one warp. The number of thread blocks is set to the number of Streaming Multiprocessors (SMs) in the system, and blocks work together processing all rows to perform each matrix multiplication from Equation 2, shown in Figure 1.

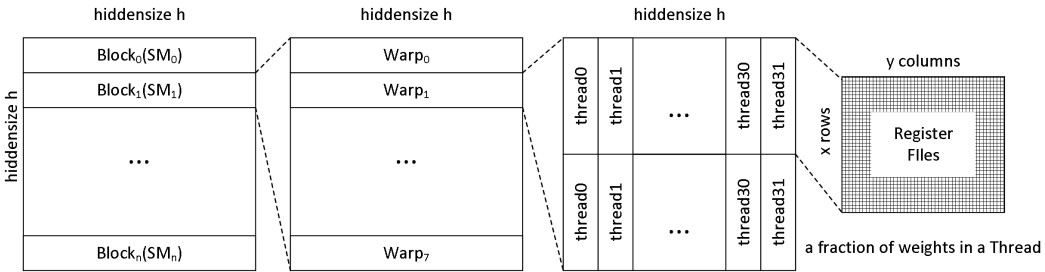

Figure 1: The mapping of work onto the GPU in a persistent approach; one row is processed by a single warp.

There are 4 stages in a persistent RNN's software pipeline: load, operate, reduce, and synchronize.

**Load:** the input activations in the current timestep are the output from previous timestep. All threads within a block cooperate to load these activations from global memory to shared memory for thread-level data re-use.

**Operate:** each thread holds $x \times y$ weights and executes $acc[i]+ = weight[i][j] * activation\_shm[j]$ computations $x \times y$ times to compute $x$ accumulations. In particular, $acc[x]$ and $weight[x][y]$ are stored in registers, and $activation\_shm[hiddensize]$ are stored in shared memory.

**Reduce:** after each thread in a warp finishes accumulating, threads belonging to the same row perform a final accumulation through shared memory and output the final result to global memory.

**Synchronize:** all threads in different blocks are synchronized by a global barrier.

Overall performance is largely dominated by math throughput in the operate stage. Since the addresses of 32 activations loaded by a warp are contiguous, there is no divergence in the shared memory load. Similarly, in one thread, weights in different rows of the same column are used by a single activation, so one thread can re-use this activation for many rows to amortize the shared memory load cost. To meet FMA and shared memory throughput, one element loaded from shared memory should do at least 4 FMA operations. So the best practice is that each thread holds more than 4 rows' weights for one column, which means one activation is re-used more than 4 times by current thread. In this way, the math units are fully utilized and not starved by memory.

### 3.3 SPARSE PERSISTENT RNNS

We now describe our novel approach to support sparse, pruned RNNs with persistent weights. In contrast to *dense* persistent RNNs, the data format of *sparse* persistent RNNs are <index, value> pairs. One <index, value> pair represents the location and value of one nonzero weight, and each thread keeps a fraction of the total nonzero weights. Additionally, in *sparse* persistent RNNs, all of the nonzero elements in one thread must belong to the same row, and the number of thread blocks depends on the hidden layer size and degree of sparsity, rather than being fixed to the number of SMs available on the chip, as in the dense case. As before, the RNN's weights are shared over all timesteps, so <index, value> pairs are stored in on-chip storage to avoid re-loading weights each timestep. Finally, there are again 4 stages in a sparse persistent RNN's pipeline:

**Load:** same as the **load** stage of dense persistent RNNs.

**Operate:** each thread holds $n$ nonzero weights ($n$ <index, value> pairs) and executes $acc+ = value[i] * activation\_in\_shm[index[i]]$ computations $n$ times in series.

**Reduce:** within each block, after each thread finishes operating and stores its accumulation into shared memory, several threads work together to generate one result for a row. All the blocks work together to process all rows.

**Synchronize:** same as the **synchronize** stage of dense persistent RNNs.

Overall performance is limited by shared memory throughput. Since the layout of the densly-packed sparse matrix doesn't match that of the original dense matrix anymore, the nonzeros in different rows of the same storage column (as opposed to logical or layout column) require different activations. This is decided by each weight's index, which means each thread cannot be guaranteed to re-use activations for nonzero weights, even though these nonzero weights may be located in the same column. Futher, the 32 unique indices in a warp may point to 32 different shared memory locations, which can lead to a maximum of 32 bank conflicts in shared memory. So, the performance of the operate stage is limited by shared memory rather than math unit, and shared memory bank conflicts become the main challenge for a sparse persistent RNN's performance.

## 4    OPTIMIZATIONS

In this section, we identify and provide solutions for several shortcomings of the previously-described implementation. The benefits of these optimizations are shown in Table 1.

First, we note that straightforwardly pruning a network does not consider the distribution of the induced zeros. The number of nonzero weights in different rows can vary, and these arbitrary distributions cause divergence and load imbalance. So, a baseline implementation must pad rows with fewer than the maximum number of nonzero weights with <index, 0> pairs, forcing all rows to have the same number of weights and, implicitly, the same workload. This technique wastes roughly 20% of the useful registers on unneeded zeros, but it still gives a performance improvement. The behavior of the network does not change since this re-introduces pruned weights with a value of 0.

### 4.1    BANK CONFLICTS

To achieve high bandwidth, shared memory is divided into 32 banks that can each service one request per cycle. With perfect request alignment, shared memory can achieve 32 requests per cycle, but multiple column indices landing in the same bank cause conflicts. These bank conflicts are serialized and are the main bottleneck of our sparse persistent RNN implementation, since in the operate stage, the 32 indices of a warp may access 32 different shared memory locations, leading to up to 32 bank conflicts. In this section, we introduce two methods to mitigate this inefficiency: wide memory loads and bank-aware weight layout. After applying these two optimizations, the total shared memory bank conflicts can be reduced by more than 80%.

**Wide Memory Loads:** In a *dense* persistent RNN, one activation can be reused by many weights in the same column. Weight reuse across samples can occur if the input to the network is a minibatch of size larger than one. Similarly, in a *sparse* persistent RNN, we can batch 4 activations from different samples together and process them at a same time(minibatch size = 4). These 4 activations can be loaded from shared memory at once by one ld.shared.v4 instruction (NVIDIA, 2017). As the addresses of 4 activations are contiguous, they belong to 4 different shared memory banks. So, the total bank conflicts can be reduced to at most $1/4$. As activations are stored in shared memory, the side-effect of using this wide memory load is that it consumes $4\times$ the shared memory. Since max hidden layer size is limited by the shared memory size, $4\times$ shared memory usage means that the maximum hidden layer size is reduced to $1/4$. So for large hidden layers, we can use ld.shared.v2 instead. This only guarantees a bank conflict reduction of $2\times$, but it only incurs a $2\times$ storage overhead. This technique introduces a trade-off between efficiency and maximum supported hidden layer size.

**Bank-Aware Weight Layout:** Each thread holds $n$ <index, value> pairs. In the operate stage, each thread executes $acc+ = value[i] * activation\_in\_shm[index[i]]$ operations one by one. Since both the activations and weight value use the same index, $i$, reordering nonzeros' locations for each row does not affect the final result, but it *can* change the access sequence to shared memory. We can construct a better shared memory access sequence to reduce bank conflicts. Performing this reordering only has to happen when the network's sparsity pattern changes. For inference, this is exactly once, and its benefit can be used for the whole lifetime of the network. Or, if training a sparse network with modifications to the pruned weight distribution over time (Guo et al., 2016; Narang et al., 2017; Zhu & Gupta, 2017), this cost can be amortized over all timesteps in the network for however many training iterations the sparsity pattern is constant. Listing 1 in Appendix A shows a greedy algorithm to generate a weight layout that reduces the amount of shared memory bank conflicts.

## 4.2 SYNCHRONIZATION

We considered two options for ensuring correct ordering between work from different thread blocks: global synchronization and **Lamport Timestamps** (Lamport, 1974). Lamport timestamps add a flag for each output value used to indicate whether or not the value has been computed. So, the load stage can make partial progress as values are marked complete, allowing the operate stage to proceed before all values are finished updating from the previous time step, as is required by global barriers. A side effect of using Lamport timestamps is that it allows a software pipeline to be used over time steps: as we process iteration $n$, we can load the states for iteration $n + 1$. Later, in iteration $n + 1$, we need to make sure the flags loaded by this prefetch indicate valid data; this can sometimes require another load for some values.

A basic implementation of Lamport timestamps uses one extra flag for every output value. This, coupled with the software pipeline, quadruples the used buffer size: $2\times$ the data for the flags, and another $2\times$ for the two concurrent pipeline stages. Our optimized version removes the flags by initializing the output buffers for each time step to -0.0f. Once each element is updated, it is implicitly valid by virtue of having a non negative zero value. (We convert all -0.0f result values to +0.0f to avoid aliasing.) So, our final implementation only doubles the memory requirements.

Each method has its own advantages and disadvantages. For example, a global synchronization requires several memory round trips to implement, but only needs to be called once per timestep. Lamport timestamps ideally require no extra memory movement, but do require multiple checks to ensure the loaded values are current. Also, to overlap the load and operate stages by preloading activations requires double-buffering, so the shared memory usage is doubled and maximum effective layer size is halved. Thus, we expose another tradeoff: larger layer sizes can be accommodated with global barriers due to the reduced shared memory usage. We found Lamport timestamps to be faster in our experiments, except for *very* large layer sizes ($>5760$).

Table 1: A naïve implementation has limited performance; our optimizations are necessary to achieve good results. (Layer size = 1152, batch size = 4, density = 10%, #timesteps = 256.)

| Configuration | Speedup (vs. dense GEMM) | Bank Conflict Penalty |
|---|---|---|
| Naïve | $2.53\times$ | 1.3 |
| Wide Memory Load | $4.28\times$ | 1.0 |
| Bank-Aware Layout | $4.56\times$ | 0.3 |
| Lamport Timestamps | $5.44\times$ | 0.3 |

## 5 EXPERIMENTS

In this section, we describe the setup and experiments performed to show the benefits of our sparse persistent RNN technique.

### 5.1 WORKLOADS

**Sparsity:** Recent efforts in pruning recurrent networks result in varying degrees of sparsity. Han et al. (2016a) were able to prune 90% of the weights in a speech recognition LSTM. Similarly, the recurrent layers of NeuralTalk (Karpathy & Li, 2014), a caption-generating network, have been pruned to around 10% density (Han et al., 2016b) without loss of accuracy. A different way to use pruning is to start with a model that is "too large" for the target hardware and prune it down to size. In this way, it is possible to produce a network with the same effective number of parameters (nonzero coefficients) that represents the sparse version of a much larger network. This has been shown to be very fruitful for RNNs (Narang et al., 2017; Zhu & Gupta, 2017) with sparsities of up to 95% for RNNs and GRUs. So, we will focus on sparsities around this common 90% target, from 80-99% sparse.

**Network size:** Common sizes of today's RNN hidden layers are 1024 to 3072. We also include larger layer sizes since 1) larger sparse networks have been shown to outperform smaller dense networks with similar capacities (Narang et al., 2017; Zhu & Gupta, 2017), and 2) network sizes in practice have historically increased as improvements in processing hardware made them feasible (Gray et al., 2017).

**Batch size:** For deployment on edge devices, smaller batch sizes of 1 to 4 are common. In data centers, inference batch sizes may be larger, and batches during training can be several hundred inputs large. We focus on the lower end, as pruning is most commonly used for deployment.

**Timesteps:** The number of timesteps used by the network depends on the use case. Translation tasks can use on the order of 10 time steps (context around the word being translated), but speech recognition may use hundreds of timesteps (audio samples). So, we explore a wide range of timesteps.

## 5.2 ALGORITHMS

To show the benefits of our sparse persistent approach, we compare against three other algorithms that can target a pruned RNN: dense GEMM (cuDNN 7.0.3), sparse GEMM (spMM in cuSparse 9.0), and a dense persistent approach (cuDNN 7.0.3). Note that the persistent approaches will not be applicable to all layer sizes due to resource constraints, with the dense persistent kernels suffering more quickly than our sparse approach. Our sparse persistent code is compiled in CUDA 9.0, and all tests are run on a NVIDIA Tesla V100.

## 5.3 RESULTS

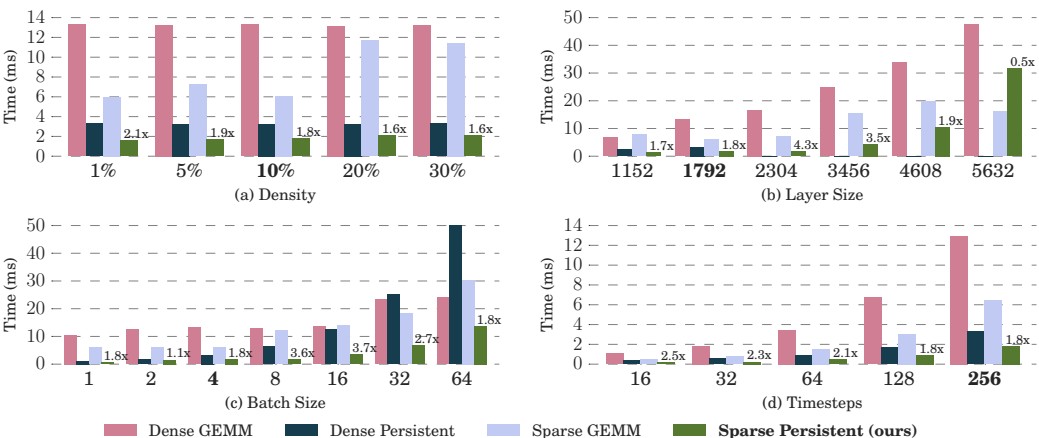

Figure 2: Different algorithms processing a pruned recurrent layer with a variety of workloads with the following baseline configuration: density of 10%, layer size of 1792, batch size of 4, and 256 timesteps (**emphasized** in each subplot). Subplot a) varies the density, b) varies the layer size, c) varies batch size, and d) varies the number of timesteps. Annotated values show the speedup of our technique over the next-best algorithm.

We use a layer size of 1796, density of 10%, batch size of 4, and 256 time steps as a baseline for Figure 2. Figure 2 (a) shows the effect of varying the density of the network. For a density of 10%, a dense persistent RNN achieves a $4.1\times$ speedup over dense GEMM, while a *sparse* persistent RNN can achieve a $7.3\times$ speedup over the same dense GEMM. When the density decreases to 1%, our approach achieves a $15.0\times$ speedup over dense GEMM.

Figure 2 (b) shows the effect of layer size on performance. Dense persistent RNNs fail after a layer size of 1792 due to insufficient registers, and the same is true for sparse persistent RNNs and a layer size of 5632 at 10% density. As the layer size grows, increased pressure on shared memory and the necessary switch to using global barriers for synchronization limit performance.

Figure 2 (c) shows how the batch size affects performance. Dense GEMM cannot fully utilize the GPU with a small batch size, so the other algorithms all outperform this simple approach. Our technique scales well to larger batch sizes, as opposed to dense persistent kernels.

Figure 2 (d) shows that the number of timesteps does not significantly affect relative performance.

A larger network with the same number of nonzero parameters is expected to outperform the smaller, dense network (Narang et al., 2017; Zhu & Gupta, 2017; Gray et al., 2017). So, we perform two experiments to show how our sparse persistent RNN technique can take advantage of this observation:

First, we vary the density of two large layer sizes, showing very large models resident on the GPU. In Figure 3 (a), we show a layer size of 2304 and vary the density. For a density of 10%, our sparse persistent RNN achieves a $9.9\times$ speedup over dense GEMM. Even at higher densities, we can improve performance by more than a factor of the sparsity by allowing a persistent approach to be used. Pushing the model size further, Figure 3 (b) shows our sparse persistent RNN approach allowing for an extremely large network with a high sparsity. For a density of 1% and a layer size of 5760, our sparse persistent RNN can achieve a $10.6\times$ speedup over dense GEMM. At 5% density, though, the overhead of the loading phase outweighs benefits of a persistent approach, and a sparse GEMM becomes faster.

Second, we fix the number of nonzero parameters present in the layer (1.32 million) and increase the layer size from 2304 (25% dense) to 11520 (1% dense) in Figure 4. As in prior work that explored this change in workload (Narang et al., 2017; Gray et al., 2017), a sparse GEMM can outperform a dense GEMM. Our sparse persistent approach can improve performance even further.

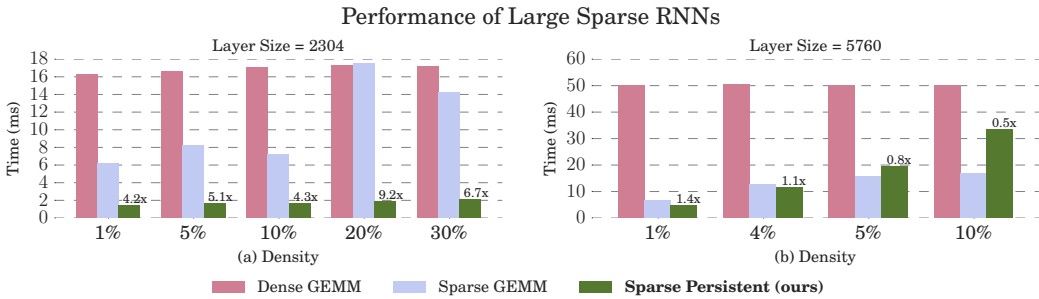

Figure 3: Relative performance of algorithms processing a pruned RNN with a large layers. Common parameters are: batch size=4, timesteps=256.

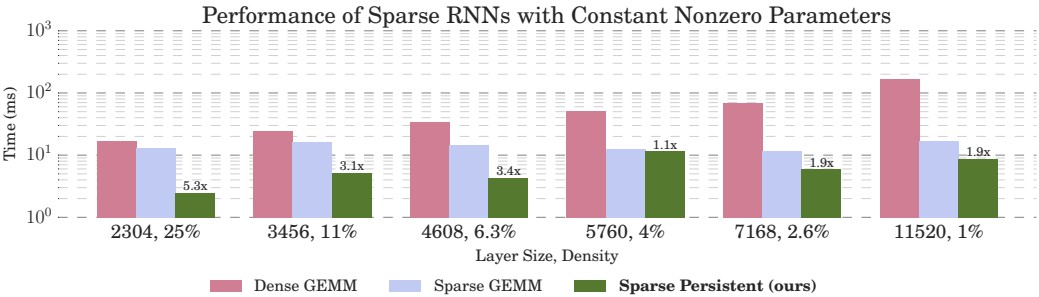

Figure 4: Various network sizes with a fixed number of nonzero parameters; i.e., larger layers are sparser. Note the log scale on the vertical axis.

## 5.4 DISCUSSION

Sparse persistent RNNs increase the performance of pruned networks, but without a series of optimizations we described in Section 4, the improvement is much lower. An optimized implementation is critical to achieve peak performance. After these optimizations, we see that our approach works best for sparser workloads, and, for most layer sizes, 10% density is sufficient to beat all other approaches, though for smaller layer sizes (up to 2304), our approach is the winner even up to a density of 30%. Different batch sizes can affect the efficiency of all techniques; in general, dense GEMMs and our approach scale the best, which tempers the benefit offered by other algorithms. Finally, the number of timesteps is largely immaterial to the relative performance of the various techniques.

We have described an efficient algorithm for recurrent layers. To exploit this algorithm, one must first prune the recurrent layers of a target network. We perform this pruning during the training process for a nureal machine translation network (Klein et al., 2017), and we look to past work (Narang et al., 2017) that has pruned a commercial speech recognition network, Deep Speech 2 (Amodei et al., 2015). The results on these real workloads are promising: for a given accuracy, we show a speedup of between $2.0\times$ to $3.1\times$ in the machine translation network's recurrent layers pruned with load-balancing in mind, and a speedup of $1.2\times$ to 2.9 on Deep Speech 2's recurrent layers pruned without load-balancing. Please refer to Appendix B for more details about the training processes and a full discussion of the results.

## 6 FUTURE WORK

Currently, we use two 32-bit registers to store a <column_index, value> pair. However, most layers are not large enough to require more than 16 bits of index data, so two column indices can be compressed into a 32-bit register, freeing up registers for more nonzeros. Similarly, we could use a lower-precision data type such as fp16 for the weights themselves to achieve the same result.

The current maximum layer size for our approach on a V100 GPU is 11520, shown in Figure 4; this limit is imposed by the 96KB shared memory usage per block. To support this layer size, we must swap Lamport timestamps for a global barrier to halve shared memory use and double the maximum layer size. The bottleneck for 1% density and a layer size of 11520 is loading activations into shared memory, rather than the operate stage. If we were to split one row among multiple blocks and each block only required a fraction of the output activations, the further reduced shared memory useage would allow for large layers to be more efficient. Using a lower-precision data type for the activations would remove the shared memory bandwidth and storage burden and achieve a similar result. Our work can be extended to multiple GPUs, allowing for larger layer sizes and more nonzero parameters.

Load balancing (due to non-uniform sparsity) is handled today with zero-padding, though several other approaches exist. Without altering the network, we can define a number of classes for different amounts of sparsity and handle each class separately, assigning each row to one class or another based on its sparsity. If we have some control over the network itself, Han et al. (2016a) have shown that load-balance aware pruning is an option, or the rows with fewer than the maximum number of nonzeros could have nonzero values re-instated to improve network accuracy ( Han et al. (2017)) with no impact to performance. Without considering accuracy, we found that for a 25% dense layer of size 2304, batch size of 4, and 256 timesteps with both straightforward (unbalanced) and load-balanced pruning, and no code modifications to take advantage of any regularity, sparse GEMM's performance does not change with load-balancing, but our approach's benefit over a dense GEMM increases from $1.14\times$ to $1.94\times$. Accuracy and performance results with this technique applied to a machine translation network are in the appendix, but the other load-balancing approaches remain future work.

## 7 CONCLUSION

We introduced sparse persistent RNNs, an efficient new algorithm for accelerating pruned recurrent networks. Further, we explored several optimizations that were needed to achieve these results on V100, a recent GPU. Our optimized technique allows for $7.3\times$, $3.4\times$, and $1.8\times$ speedups against dense GEMM, sparse GEMM, and dense persistent implementations for a hidden layer of size 1792, batch size of 4, and a density of 10%. We show that much larger networks can be deployed onto a GPU of a fixed size with performance increases of around $5\times$ over the next best solution for a density of 1-30% on a layer size of 2304; notably, this sparsity range includes denser workloads than typically perform worse with sparse optimizations. We also show promising results on much larger layers - 7168 and 11520 achieve speedups of $1.9\times$ for 2.6% and 1% densities, respectively. Our approach speeds up pruned NMT and speech recognition networks' recurrent layers by up to $3\times$. Finally, load-balanced pruning can significantly improve a network's throughput, and our technique is necessary to achieve both high performance and accuracy in some recurrent layers, as detailed in the case studies in our appendix.

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

## APPENDIX A: ALGORITHM FOR BANK-AWARE WEIGHT LAYOUT

As discussed in Section 4, we decouple the physical weight layout from the logical view. Here, we share the greedy algorithm we used, which is one possible approach.

---

**Algorithm 1:** Optimize a row of nonzero weights to minimize bank conflicts

---

*Invoked for each row independently*
**Input**   : $Row_X(0..N-1)$ where $Row_X(i) = <index_i, value_i>$
**Initialization:** $Color(0..31)$ *with (X%32 .. (X+31)%32)*
**Output :** $Row_X^{aware}(0..N-1)$

---

**for** *<index,value> pair in $Row_X$* **do**
  $index\_t = index$ ;
  $bank = index\_t\%32$ ;                                    ▷ find its bank
  **while** $Color(bank) > N$ **do**
    $index\_t + +$ ;    ▷ Color(bank) is occupied.  Use a unfilled Color
    $bank = index\_t\%32$ ;
  **end**
  $Row_X^{aware}(Color(bank)) = <index, value>$ ;
  $Color(bank) = Color(bank) + 32$ ;    ▷ Update Color(bank)'s next location
**end**

---

## APPENDIX B: CASE STUDIES

Having described our efficient algorithm for accelerating pruned recurrent layers, we now perform case studies showing its utility, as well as its generality. The results in the main body are for RNNs only, but in this first section, we show performance results for LSTM (Hochreiter & Schmidhuber, 1997) layers. Implementation differences are fairly straightforward. Instead of one gate in an RNN, each thread must be responsible for four gates in an LSTM. As a result, each thread's weights belong to one of four rows of the original weight matrix, and each row requires a different activation function. The increased gate count results in an increased number of weights for the same layer size. Dense persistent approaches will not be available for an LSTM of layer size of 1024 as a result (on the V100 GPU used for the experiments in this section).

We first explore pruning a neural machine translation network to see how accuracy changes with sparsity. More importantly, we care about execution speed and accuracy; sparsity is merely one way to trade off speed for accuracy. (Layer size is much more straightforward.) We outline our simple pruning and training procedures here. Next, we use results from prior work that pruned a productized speech recognition network. We show that our technique allows for higher performance for a given accuracy; without our technique, sparsity may not lead to any gains whatsoever.

### 1. NEURAL MACHINE TRANSLATION

### A. SETUP

We use OpenNMT (Klein et al., 2017) to perform translation from English to German using the WMT15 data set as our training data and the newstest2013 data set for validation. The common network architecture is a 2-layer (for both encoder and decoders) LSTM; our only variation from network to network is the layer size. This can be reproduced by following the excellent tutorial. We trained each network with only two GPUs. Performance results were gathered on a single NVIDIA Tesla V100 accelerator.

### B. PRUNING STRATEGIES

Each of our pruned networks use a magnitude-based pruning scheme, in which we consider only the magnitude of the weight when deciding which weights to prune. See et al. (2016) found that different

pruning methods affect accuracy differently, and we look at two options: naïve and row-balanced. In both cases, each layer of the network is pruned to the same target density.

**Naïve pruning** does not consider where the weight is within a layer; that is, all gates and all rows are considered to be the same. The effect of this strategy, as noted in past work, is that the forget gate is much sparser than the other gates in an LSTM; intuitively, the relative lack of constraints on the sparsity should be less of a burden on the network's convergence.

**Row-balanced pruning** has the constraint that each row in a layer must have the same number of nonzero values. The effect of this strategy is that the weights are naturally more load-balanced for use with our approach, which should allow for higher performance (as evidenced in Section 6).

When examining the effective density of the naïve-pruned layers after padding with zeros for load-balancing (see Section 4), we found that the amount of remaining sparsity was not sufficient to exploit with a sparse method[1]. Further, the difference between the network accuracy of the two pruning options was negligible: less than 0.1 BLUE points in each case. So, we focus only on row-balanced pruning.

## C. TRAINING PROCESS

For all networks, we train for the default 13 epochs, with all other hyperparameters unchanged from the default.

Sparse techniques are not limited to inference; recent work has shown that training with pruned weights is a viable option (Guo et al., 2016; Narang et al., 2017; Zhu & Gupta, 2017). For our experiments, we adopt a simple methodology: after one-half epoch of training, we prune immediately to the target density. Like Guo et al. (2016), we continue to *update* a "master copy" of un-pruned parameters during the backwards pass, but the pruned weights are used for computation during the forwards and backwards passes. As all the weights are updated, the weights pruned during one pruning step may be re-introduced in the following pruning step, changing the sparsity pattern between pruning phases.

This pruning step could impose a large overhead, so we only prune every half-epoch for a total of 21 pruning steps. Thus, the first half epoch is fully dense, the sparsity pattern changes every half-epoch through the end of epoch 11, and epochs 12 and 13 fine-tune the final sparsity pattern.

## D. ACCURACY RESULTS

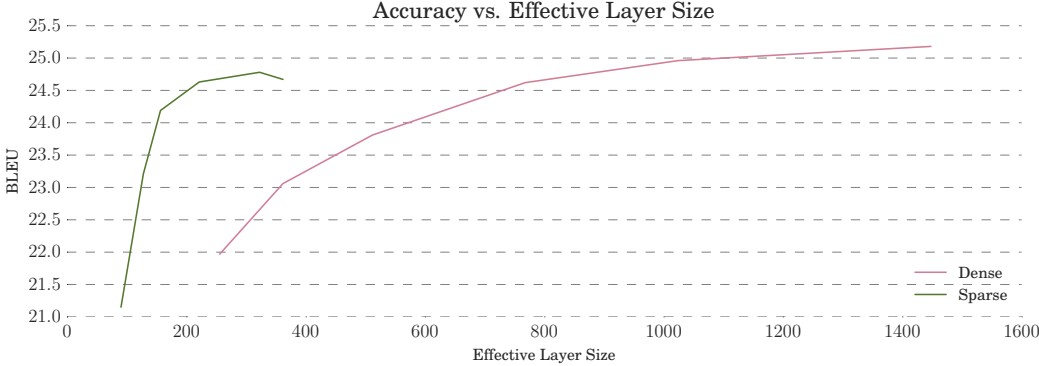

Figure 5: Training with sparsity can yield a higher accuracy for a given number of nonzero parameters ("Effective Layer Size").

Figure 5 shows the accuracy of various networks trained with and without sparsity, and the first four columns of Table 2 show the detailed results. As expected, network accuracy increases with layer size, and, as demonstrated in other work, a large and sparse network can outperform a smaller dense network with the same number of nonzero parameters ("Effective Layer Size"). One interesting result

---

[1]This is not necessarily the case when considering each gate separately, or for vanilla RNNs.

suggests the scalability of these results: a larger, sparser network (layer size of 1448 at 4.7% density for an effective size of 322) out-performed a slightly-smaller, not-as-sparse network (layer size of 1024 at 12.5% density for an effective size of 362). This is why the upper tail of the "sparse" curve in Figure 5 dips down; in truth, the method should continue to scale. Omitting the 1448 layer size result would show monotonically increasing accuracy, as would ordering the accuracies by underlying (unpruned) layer sizes for these densities, as shown in Table 2.

Even with our relatively simple training procedure, network accuracy was within a reasonable distance of the dense baseline, only 0.4 BLEU worse at the largest (and most accurate) configuration. We note that these accuracy results were achieved with a very simple method and can likely be improved by:

- More elaborate pruning schedules during training

- Performing a sensitivity analysis and pruning layers to different densities

- Pruning gates within a layer to different densities

- Pruning only the recurrent weights, rather than the recurrent *and* feed-forward weights

- Adjusting hyperparameters, such as dropout

- Performing more fine-tuning of the final network (at the cost of extra training time)

Table 2: BLEU Scores and Execution Times of Various Configurations

| Network | Layer Size | Density | Eff. Size | BLEU | ms (GEMM) | ms (Persistent) |
|---------|-----------|---------|-----------|------|-----------|-----------------|
| Dense | 256 | 100% | 256 | 21.97 | 1.74 | 1.11 |
|  | 362 | 100% | 362 | 23.06 | 2.05 | 1.14 |
|  | 512 | 100% | 512 | 23.81 | 3.00 | 1.12 |
|  | 768 | 100% | 768 | 24.62 | 3.55 | 1.26 |
|  | 1024 | 100% | 1024 | 24.96 | 4.34 | 3.65 |
|  | 1448 | 100% | 1448 | 25.18 | 7.21 | – |
| Sparse | 256 | 12.5% | 90 | 21.15 | 2.44 | **0.33** |
|  | 512 | 6.25% | 128 | 23.21 | 2.21 | **0.37** |
|  | 768 | 4.17% | 156 | 24.19 | 2.19 | **0.48** |
|  | 1024 | 4.7% | 222 | 24.60 | 1.87 | **0.55** |
|  | 1024 | 12.5% | 362 | 24.67 | 3.19 | **0.63** |
|  | 1448 | 4.7% | 322 | 24.78 | 3.39 | **0.78** |

## E. PERFORMANCE RESULTS

While these accuracy results are not necessarily a surprise, a missing part of most treatments of this behavior is the throughput of the network on a given architecture. To show that a large, sparse network is not only a good tradeoff for accuracy but also for performance, we compare different layers' performance with all state-of-the-art algorithms that support each particular layer size. In particular, we use a dense GEMM (cuBLAS), dense persistent GEMM (cuDNN), sparse GEMM (cuSPARSE), and our sparse persistent GEMM. We profile these LSTM recurrent kernels on an NVIDIA V100 GPU. An important note is that this gives us access to 96KB of shared memory per thread block, in turn allowing for larger layer sizes to be supported more efficiently. Table 2 has the details for each layer size; our approach's performance on the sparse network is given with **emphasis**. Note that these absolute improvements are for a single layer; a production network would be composed of multiple layers. More importantly, the relative improvement can lead to a proportional speedup in training time, which can be days or weeks for large networks.

Our findings in Figure 6 show that while a pruned version of a network can run significantly faster than the dense version of that same network size (see our main text), the same accuracy could be obtained with a smaller, dense network. If this small, dense network is able to be implemented with a persistent kernel, then it can be more performant – without our technique. cuSPARSE is very effective at outperforming dense GEMM at network sizes that a dense persistent approach cannot handle, but our technique pushes this sparse/dense crossover point much lower. Now, sparse networks can outperform dense ones, even at smaller effective layer sizes.

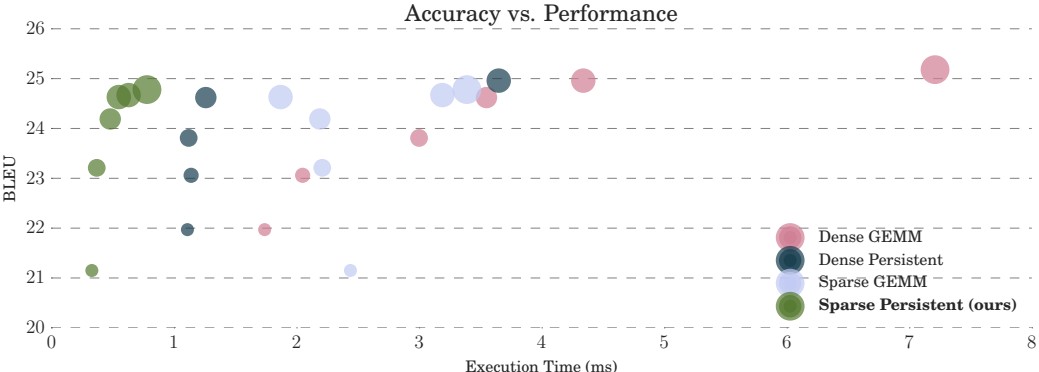

Figure 6: Our efficient algorithm can be used on pruned workloads; for a given performance target, pruned networks using a sparse persistent approach provide the best BLEU score. Likewise, for a given network accuracy, a pruned network accelerated with our algorithm gives the highest throughput. The size of each marker is proportional to the size of the underlying (unpruned) hidden layer.

## 2. SPEECH RECOGNITION

Deep Speech 2 (Amodei et al., 2015) is a state-of-the-art speech recognition network that has been pruned to expose an accuracy/sparsity tradeoff (Narang et al., 2017), so we refer to these resources for details regarding the network and associated training and pruning approaches. Our purpose here is to see the performance of various algorithms on the different versions of the network that they trained. Since we did not train these networks ourselves, we generated a random, unstructured nonzero pattern for each workload; no load-balancing has been done prior to the optimization steps of our algorithm. We reproduce the authors' information from Table 3 for layer size, sparsity, and accuracy (character error rate, CER) in our own Table 3, adding the performance of different algorithms on V100 for a batch size of 1 and 256 time steps. (As shown above, varying time steps does not significantly change the relative performance. We discuss the batch size further below.)

Table 3: CER and Execution Times of Various Deep Speech 2 Networks

| Network | Layer Size | Density | Eff. Size | CER | ms (GEMM) | ms (Persistent) |
|---------|-----------|---------|-----------|-------|-----------|-----------------|
|         | 704       | 100%    | 704       | 14.50 | 3.02      | 0.74            |
| Dense   | 1760      | 100%    | 1760      | 10.67 | 10.80     | 1.11            |
|         | 2560      | 100%    | 2560      | 9.43  | 14.30     | –               |
|         | 1760      | 12%     | 610       | 12.88 | 3.95      | **0.72**        |
| Sparse  | 2560      | 12%     | 887       | 10.59 | 4.70      | **0.89**        |
|         | 3072      | 12%     | 1064      | 10.25 | 6.09      | **0.94**        |

We see the familiar pattern: for a given effective layer size, larger and sparse layers lead to more accurate models than small and dense layers. However, Figure 7 reveals that the dense persistent approach is along the pareto-optimal curve *in the absence of our technique*. In other words, sparse methods, though they may be faster than a dense GEMM (especially for a batch size of 1), can not out-perform a dense persistent approach. If speed is a metric of importance, then dense persistent kernels are the answer. If accuracy is the most important metric, then past work has *not* shown that a sparse network can outperform any dense network.

Including our sparse persistent kernels, however, shows that they are better than dense persistent kernels for a given network; for a particular speed target, they can provide higher accuracy with a pruned network. At very low batch sizes, a sparse GEMM can occupy a useful spot in the accuracy/speed tradeoff curve, but with only a batch size of 8, they fall to last place at all instances of the network. Thus, in the absence of our technique, there would be no reason to prune a network batched inference performance.

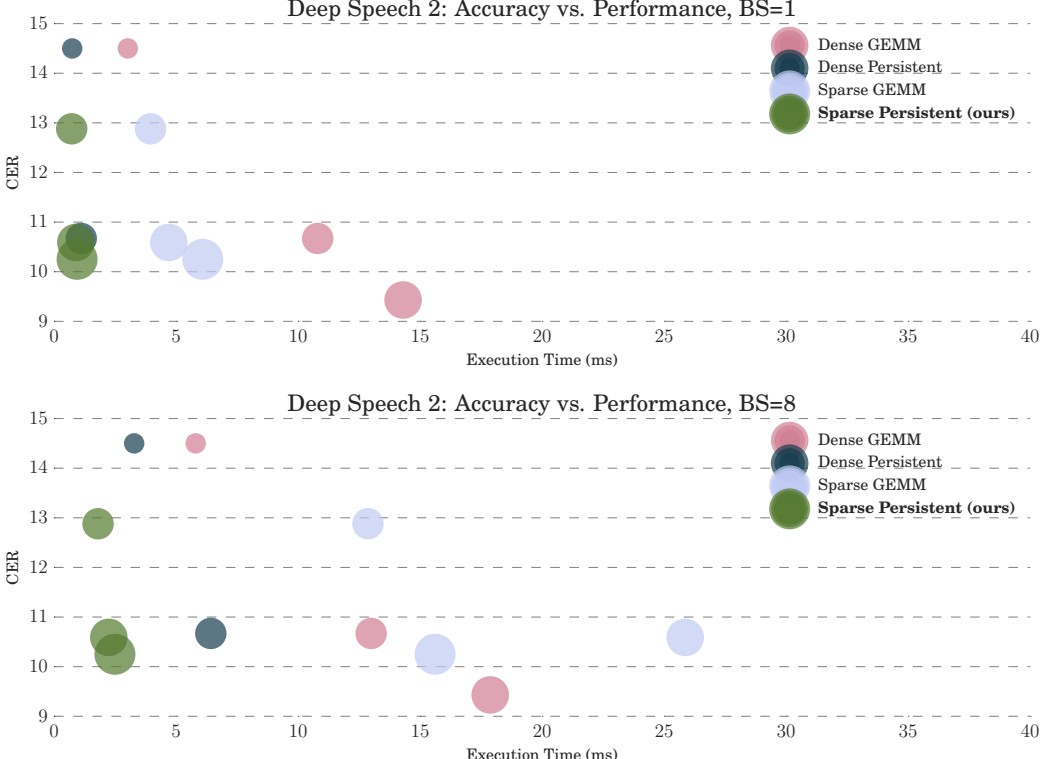

Figure 7: Various algorithms applied to different Deep Speech 2 models; for an aggressive performance target, pruned networks using a sparse persistent approach provide the lowest error rate. Likewise, for all but the most stringent error rates, a pruned network accelerated with our algorithm gives the highest throughput. The size of each marker is proportional to the size of the underlying (unpruned) hidden layer.

## 3. CONCLUSIONS

Without consideration of the accuracy of pruned networks and their execution speed (on state-of-the-art algorithms) together, the conclusion that a large, sparse network is better than a small, dense one is not sufficiently proven. Our technique and these case studies show, among these insights, that this *can* be the case, however:

- Our technique extends to LSTMs with little effort.
- Our technique allows for pruned recurrent layers to run more efficiently than with any other existing algorithm.
- Naïve pruning is not necessarily better than pruning with load-balancing in mind, especially when considering achievable performance.
- Like past work, we see that a larger, sparse network can be more accurate than a smaller, dense one.
- Comparisons against persistent kernels, which can beat non-resident sparse approaches, show that for a given accuracy or performance target, pruning a network and using our technique is the best choice.

