# OpenReview forum: "Sparse Persistent RNNs: Squeezing Large Recurrent Networks On-Chip"
_ICLR.cc/2018/Conference — Accept (Poster)_

### Official Review · AnonReviewer2 · 2017-11-22
**The paper devises sparse GPU kernels for RNNs**

**Rating:** 6
**Confidence:** 2

**Review:**

The paper devises a sparse kernel for RNNs which is urgently needed because current GPU deep learning libraries (e.g., CuDNN) cannot exploit sparsity when it is presented and because a number of works have proposed to sparsify/prune RNNs so as to be able to run on devices with limited compute power (e.g., smartphones). Unfortunately, due to the low-level and GPU specific nature of the work, I would think that this work will be better critiqued in a more GPU-centric conference. Another concern is that while experiments are provided to demonstrate the speedups achieved by exploiting sparsity, these are not contrasted by presenting the loss in accuracy caused by introducing sparsity (in the main portion of the paper). It may be the case by reducing density to 1% we can speedup by N fold but this observation may not have any value if the accuracy becomes  abysmal.

Pros:
- Addresses an urgent and timely issue of devising sparse kernels for RNNs on GPUs
- Experiments show that the kernel can effectively exploit sparsity while utilizing GPU resources well

Cons:
- This work may be better reviewed at a more GPU-centric conference
- Experiments (in main paper) only show speedups and do not show loss of accuracy due to sparsity

---

> ### Author Response · Authors · 2017-12-28
> **Accuracy-sparsity tradeoffs in past work**
>
> Thank you for your comments and observations.  Let us first address the critical importance of network accuracy after pruning.  We completely agree that large speed improvements are a moot point if the accuracy does not hold up.  However, an exhaustive study of the sparsity/accuracy tradeoff is out of the scope of this paper.  Instead, we refer to several other published results that show good accuracy results for recurrent networks around the 10% density point [Han et al. 2016(a,b), Narang et al. 2017, See et al. 2016, Anonymous 2018].  So, we centered our experiments around this density and swept from 1% to 30% to cover a wider range.  After submission, densities down to 3% have been recently used to achieve state of the art results on some workloads (https://blog.openai.com/block-sparse-gpu-kernels/), and we show good speedups for higher densities, especially when the layer size is too large for a dense persistent kernel.  Finally, we provided more accuracy vs. sparsity vs. speed results in the appendix to show why our technique is important.  We'll gladly move this analysis into the main paper if extending beyond 8 pages is preferable to merely including references to other works with accuracy results.
>
> We feel that this work is relevant to the ICLR audience.  As you noted, sparsity is not regularly accelerated by deep learning libraries.  More importantly, as we show in our appendix, some recurrent layers are actually better off staying dense and using a persistent approach if possible (without our sparse persistent technique).  Simply increasing accuracy for the same number of effective parameters is not sufficient to claim success; the network's speed may not increase over a dense network!  Thus, one of the fundamental benefits of sparsity is tempered in some cases.  Our main contribution shifts this balance back in favor of pruning for recurrent layers.

---

### Official Review · AnonReviewer1 · 2017-11-27
**Sparse Persistent RNNs Review: Limited novelty over persistent RNNs**

**Rating:** 6
**Confidence:** 4

**Review:**

This paper introduces sparse persistent RNNs, a mechanism to add pruning to the existing work of stashing RNN weights on a chip. The paper describes the use additional mechanisms for synchronization and memory loading.

The evaluation in the main paper is largely on synthetic workloads (i.e. large layers with artificial sparsity).  With evaluation largely over layers instead of applications, I was left wondering whether there is an actual benefit on real workloads. Furthermore, the benefit over dense persistent RNNs for OpenNMT application (of absolute 0.3-0.5s over dense persistent rnns?) did not appear significant unless you can convince me otherwise.

Storing weights persistent on chip should give a sharp benefit when all weights fit on the chip. One suggestion I have to strengthen the paper is to claim that due to pruning, now you can support a larger number of methods or method configurations and to provide examples of those.

To summarize, the paper adds the ability to support pruning over persistent RNNs. However, Narang et. al., 2017 already explore this idea, although briefly. Furthermore, the gains from the sparsity appear rather limited over real applications. I would encourage the authors to put the NMT evaluation in the main paper (and perhaps add other workloads). Furthermore, a host of techniques are discussed (Lamport timestamps, memory layouts) and implementing them on GPUs is not trivial. However, these are well known and the novelty or even the experience of implementing these on GPUs should be emphasized.

---

> ### Author Response · Authors · 2017-12-28
> **Benefit over dense persistent RNNs, request for clarification**
>
> Thank you comments and suggestions.  It is fair to wonder about the performance on real workloads; we decided to show the performance of our technique over a wide range of synthetic workloads so that practitioners can look to see where their application lives in the space and judge the relative performance accordingly.  Our appendix shows the performance of recurrent layers of one particular application.
>
> With respect to the speedup over the dense persistent LSTMs in the OpenNMT network, 0.3-0.5s (looking at layers of the same size) is not the proper comparison.  Instead, we think that the comparison should be between networks of the same accuracy.  In this case, the improvement is up to 0.7ms (from 1.26ms for a BLEU score of 24.62 to 0.55ms for a BLEU score of 24.60).  Also, this is a per-layer improvement; a full network will be composed of several such layers leading to a larger absolute improvement for the network as a whole.  More important than absolute speedup for a single iteration, however, is the potential speedup for training networks.  This absolute 0.7ms reduces the run time to 44% of the previous time, roughly halving the time needed to train the network to a given accuracy.    We'll make this clear in the final text.  Finally, it's worth noting that without our contributions, the benefit of sparsity would be negative: existing sparse methods are worse than persistent kernels for a given accuracy or speed target on the workloads we studied.
>
> We have a question about your suggestion to claim support for a larger number of methods.  We do claim this: Figure 3 shows that we can support larger layers in a persistent approach than existing methods.  Please let us know if we've misunderstood; we welcome opportunities to strengthen this paper!
>
> We will certainly move the NMT evaluation into the main paper if the reviewers think it warrants the extra space.  We agree that it naturally belongs there.
>
> We're also willing to emphasize the non-trivial aspects of the optimizations we used, as opposed to the brief mention in past work you point out.  It is exactly these optimizations which take the bulk of the main paper; was there something in particular you suggest adding?

---

> > ### Comment · AnonReviewer1 · 2018-01-02
> > **response**
> >
> > When I read the paper (and not the Appendix), I was left wondering how much this benefits real applications as opposed to synthetic workloads. Figure 3 is in the right direction. But can you connect the dots for the reader and describe some applications which especially benefit from large layers of the specific sizes you have mentioned?
> >
> > I agree that the optimizations are non-trivial but if they can be made interesting to the larger ICLR community, it will be great!
> >
> > I have upgraded my score. I still find the paper little bit weak on novelty but I am confident that you will fix the other issues/clarifications raised in my review, in the final revision.

---

### Official Review · AnonReviewer3 · 2017-11-27
**Novel and impactful contributions, but unclear relevance and expected audience**

**Rating:** 6
**Confidence:** 2

**Review:**

The paper proposes improving performance of large RNNs by combing techniques of model pruning and persistent kernels. The authors further propose model-pruning optimizations which are aware of the persistent implementation.

It's not clear if the paper is relevant to the ICLR audience due to its emphasize on low-level optimization which has little insight in learning representations. The exposition in the paper is also not well-suited for people without a systems background, although I'll admit I'm mostly using myself as a proxy for the average machine learning researcher here. For instance, the authors could do more to explain Lamport Timestamps than a 1974 citation.

Modulo problems of relevance and expected audience, the paper is well-written and presents useful improvements in performance of large RNNs, and the work has potential for impact in industrial applications of RNNs. The work is clearly novel, and the contributions are clear and well-justified using experiments and ablations.

---

> ### Author Response · Authors · 2017-12-28
> **Clarifying relevance**
>
> Thank you for your time and comments.  With respect to showing the relevance to ICLR, we think the results of our work are very important.  Let us try to clarify this relevance by presenting the results of the appendix _without_ the context of the main paper: "For the recurrent layers of the network we studied, there's no need to prune the weights.  A dense persistent implementation of the network is faster for the same accuracy as a pruned network, or more accurate at a given speed target."
>
> There has been significant interest in model pruning, mostly for the purposes of increasing performance.  However, realizing increased performance often requires some type of structured pruning, such as pruning filters or channels from convolutional networks, or leaving dense blocks in recurrent networks.  (As Narang et al. noted in their 2017 work at ICLR, cuSPARSE achieves limited speedup for unstructured sparsity, especially for large batch sizes.)  However, imposing structure on the sparsity reduces the degrees of freedom; unstructured sparsity can represent a proper superset of the patterns that any structured sparse layer can represent.  Therefore, it is preferable from a model's point of view to use sparsity without any structure (if sparsity is to be used at all, and second-order regularization effects of imposed structure notwithstanding).  So, we are motivated to find the efficient method, presented in the main section, to accelerate recurrent layers with unstructured sparsity.
>
> However, presenting an efficient method is only half the story; to start filling in the pieces, we included our appendix (as an appendix, in order to stay within the suggested page limit).  We show how both accuracy and speed change with sparsity.  In particular, without our method, unstructured sparsity (preferred by the model) is inferior to a dense network.  Dense persistent kernels are faster and more accurate than their pruned cuSPARSE counterparts for the model we studied.  We will make these points more clear in the next version of the text -- as well as spending some more space on describing Lamport Timestamps!

---

### Author Response · Authors · 2018-01-05
**New revision**

Thanks to the feedback of the reviewers, we have updated our submission.  The key differences are these:
- All performance numbers are now gathered on a V100 GPU
- We added more information about Lamport timestamps in the text to clarify their behavior and benefit
- We add Deep Speech 2 to the case study in the appendix (up to a 3x speedup for baseline accuracy)
- We mention the speedups from the case study in the main text to make concrete layer speedups on real tasks clear to readers
- We make the "side effect" of our algorithm more clear: it is now worthwhile to prune recurrent layers, whereas persistent kernels before would regularly outperform sparse GEMMs for a target accuracy.  We see this point as a key contribution.

---

### Decision · Program_Chairs · 2018-01-29
**ICLR 2018 Conference Acceptance Decision**

**Decision:**

Accept (Poster)

**Comment:**

The reviewers find the work interesting and well made, but are concerned that ICLR is not the right venue for the work.  I will recommend that the paper be accepted, but ask the authors to add the NMT results to the main paper (any other non-synthetic applications they could add would be helpful).